# Efficient Method for Generating Point Mutations in the Vaccinia Virus Genome Using CRISPR/Cas9

**DOI:** 10.3390/v14071559

**Published:** 2022-07-18

**Authors:** Laetitia Boutin, Estelle Mosca, Frédéric Iseni

**Affiliations:** Virology Unit, Institut de Recherche Biomédicale des Armées, 91220 Brétigny-sur-Orge, France; laetitiaboutin.lb@gmail.com (L.B.); estelle.mosca@intradef.gouv.fr (E.M.)

**Keywords:** vaccinia virus, CRISPR-Cas9, homologous recombination, PAM sequence

## Abstract

The vaccinia virus (VACV) was previously used as a vaccine for smallpox eradication. Nowadays, recombinant VACVs are developed as vaccine platforms for infectious disease prevention and cancer treatment. The conventional method for genome editing of the VACV is based on homologous recombination, which is poorly efficient. Recently, the use of CRISPR/Cas9 technology was shown to greatly improve the speed and efficiency of the production of recombinant VACV expressing a heterologous gene. However, the ability to rapidly recover viruses bearing single nucleotide substitutions is still challenging. Notwithstanding, ongoing studies on the VACV and its interaction with the host cell could benefit from viral gene targeted mutagenesis. Here, we present a modified version of the CRISPR/Cas9 system for the rapid selection of mutant VACV carrying point mutations. For this purpose, we introduced a silent mutation into the donor gene (which will replace the wildtype gene) that serves a double function: it is located in the PAM (NGG) sequence, which is essential for Cas9 cleavage, and it alters a restriction site. This silent mutation, once introduced into the VACV genome, allows for rapid selection and screening of mutant viruses carrying a mutation of interest in the targeted gene. As a proof of concept, we produced several recombinant VACVs, with mutations in the E9L gene, upon which, phenotypic analysis was performed.

## 1. Introduction

The vaccinia virus (VACV) is a well-known poxvirus that was once successfully used as a live vaccine to eradicate variola virus. Today, the VACV is still widely used, and has been bioengineered to serve as a versatile vector for the development of vaccines against infectious diseases, as well as for cancer vaccines and oncolytic immunotherapy. Thus, several vaccine candidates against HIV and the influenza virus have been evaluated in clinical trials [1], and, very recently, a modified VACV co-expressing the SARS-CoV-2 spike and nucleocapsid antigens has advanced into phase I trials [2]. For enhanced safety, most of these vaccines have been based on an attenuated version of the Ankara vaccinia virus (MVA), of which the genome has lost approximately 30 kb of DNA after being passaged over 500 times in cell culture [3]. As a consequence, the MVA is unable to replicate in most mammalian, including human, cells [4]. On the contrary, VACV vectors genetically engineered for oncolytic virotherapy are replication competent. These modified viruses, in which specific genes have been deleted (typically J2R, encoding a thymidine kinase, or F4L, encoding a subunit of ribonucleotide reductase), can selectively infect and kill tumor cells without damaging normal tissues [5]. Oncolytic VACV armed with granulocyte/macrophage-colony-stimulating factor (GM-CSF) and a bifunctional cytosine deaminase/uracil phosphoribosyl transferase gene (FCU1) are currently being tested in clinical trials to treat colorectal and malignant solid tumors, respectively [1,6].

In addition to showing good safety levels, stable expression of exogenous antigens, and convenient storage, the VACV has other interesting features, making it relatively easy to manipulate. First, the 200-kb viral genome can accommodate insertions or deletions of DNA fragments of up to 25 kb, which allows the expression of multiple antigens and immune-modulating proteins [7]. Second, genetic recombination catalyzed by the viral DNA polymerase (E9) during infection [8] allows for easy VACV genome editing by homologous recombination [9]. However, this activity is poorly efficient (<1% for VACV), and often requires the insertion of selection markers (in addition to the gene of interest) to sort the expected recombinant viruses from parental wildtype viruses. Because the presence of selectable markers raises safety concerns if the VACV vaccine were to advance into clinical trials, recent studies aiming to genetically engineer poxviruses to rapidly generate marker-free viruses have been conducted.

The CRISPR (clustered regularly interspaced short palindromic repeats)/Cas9 system is a natural bacterial immune mechanism that has been adapted to an efficient and versatile gene editing technology to modify the genome of eukaryotic cells [10]. This system consists of the Cas9 nuclease forming a ribonucleoprotein complex with a guide RNA (gRNA) that can bind to and cut a specific DNA target in a whole genome. The induced double-strand break (DSB) is then repaired through either homologous recombination (HR) or a nonhomologous end joining (NHEJ) mechanism. The NHEJ generates insertions and deletions, leading to the genetic knockout of targeted genes, whereas HR permits gene replacement through the insertion of a homologous DNA template [10].

Recent advances in the application of CRISPR/Cas9 technology have greatly facilitated the recovery of recombinant VACV [11,12,13,14]. Indeed, combining the ability to induce DSBs at specific loci in the VACV genome, and that of the virus, to perform homologous recombination was shown to greatly accelerate the generation of VACV carrying foreign genes, and provides clear evidence that the CRISPR/Cas9 system significantly improves the selection of recombinant viruses from background parental viruses (sometimes >90%). Although the technical leap for the generation of VACV vaccines is obvious, allowing the production of marker-free viruses within two weeks [14], the selection of mutant VACVs bearing point mutations in a given gene is still technically very difficult. However, such a tool would be of great interest for structure–function studies of specific viral genes.

Here, we present a modified version of the CRISPR/Cas9 system that permits the rapid selection of mutant VACV carrying single mutations. As a proof of concept, we recovered several VACVs with lesions in the E9L gene. The phenotype of these mutant viruses was then evaluated.

## 2. Materials and Methods

### 2.1. Cells Lines and Viruses

CV-1 cells (monkey kidney fibroblast cells, ATCC CCL-70) and Vero cells (monkey kidney epithelial cells, ATCC CCL-81) were grown in Dulbecco’s modified Eagle’s medium (DMEM; GIBCO, Courtaboeuf, France) supplemented with 10% heat-inactivated fetal calf serum (FCS, Thermo Fisher Scientific), and maintained at 37 °C in a 5% CO_2_ atmosphere.

The VACV (Copenhagen strain) used in these experiments was produced by infecting 90% confluent Vero cells at a multiplicity of infection (MOI) of 0.2. The virus was titrated as described below.

### 2.2. Plasmid Vectors

The pCMV-Cas9 vector expressing the Streptococcus pyogenes Cas9 protein was purchased from Sigma-Aldrich. The nuclear localization signal (NLS) sequence (PKKKRKV) fused to the C-terminal extremity of Cas9 was removed by PCR-directed mutagenesis of pCMV-Cas9 using primers described in Table 1.

The pU6-gRNA plasmids encoding guide RNAs targeting the E9L gene (Table 2) were obtained from Sigma-Aldrich. To avoid off-target cleavage in the VACV genome, gRNA sequences were submitted to the Cas-OFFinder algorithm [15]. pCMV-Cas9ΔNLS and pU6-gRNA were amplified in DH5 α *E. coli*.

The VACV (Copenhagen strain) E9L gene was cloned into the pGEM-T vector (Promega, Charbonnières, France) with primers described in Table 1. The various mutations introduced into the E9L gene were generated by PCR-based site-directed mutagenesis (Table 1). pGEM-T-E9L constructs were amplified in JM109 cells (Promega).

### 2.3. CRISPR/Cas9-Mediated Homologous Recombination

On day 1, 5 × 10^6^ CV-1 cells were infected with VACV at an MOI of 0.02 in DMEM supplemented with 0.5% *v*/*v* FCS for 1 h at 37 °C in a 5% CO_2_ atmosphere. After infection, cells were electroporated using the Neon Transfection System (Thermo Fisher Scientific, Courtaboeuf, France) as follows: cells were washed once in 1X phosphate-buffered saline (PBS), harvested by trypsinization, and resuspended in 100 µL electroporation R buffer (1 × 10^7^ cells/mL). Cells (100 µL) were mixed with 2.25 µg of each plasmid: pCMV-Cas9ΔNLS, the pGEM-T-E9L vector, and the pU6-gRNA of interest. In control experiments (i.e., transfection without pCMV-Cas9ΔNLS and pU6-gRNA), 4.5 µg of empty pmaxGFP plasmid (Lonza, Basel, Switzerland) was co-transfected together with the pGEM-T-E9L vector. The cell/DNA mixture was aspirated into a 100 µL Neon tip and submitted to two electric pulses at 1050 volts for 30 ms. Cells were then seeded in six-well plates containing warm DMEM supplemented with 10% FCS and incubated for 3 days at 37 °C and 5% CO_2_.

### 2.4. Screening of Recombinant Viruses

After a single freeze–thaw cycle, the viral suspension was recovered and diluted in DMEM before infection of Vero cells seeded in six-well plates. After 1 h of adsorption, the residual inoculum was removed and replaced with DMEM supplemented with 10% FCS and 1% agarose. The plates were incubated at 37 °C and 5% CO_2_. At 3 days post-infection, 10 to 20 individual plaques were picked and used to infect Vero cells seeded in 96-well plates for 3 days. For each clone, the viral DNA genome was extracted using the QIAamp DNA mini kit (Qiagen, Courtaboeuf, France). Specific genomic regions of the E9L gene were PCR-amplified using primers described in Table 1. PCR amplicons were purified using a QIAquick purification PCR kit (Qiagen), and digested with BspEI or MmeI restriction enzymes (NEB, Evry, France). Digested products were separated by 0.8% agarose gel electrophoresis and visualized with ethidium bromide. Mutant viruses were further characterized by Sanger sequencing of the E9L gene to ensure the presence of the expected mutations. Samples for sequencing were prepared as follows: after VACV genome extraction, E9L was PCR-amplified using Q5(R) High-Fidelity DNA Polymerase (NEB) and the primers E9-seq-F and E9-seq-R (Table 1). PCR products were purified with the QIAquick PCR purification kit. Samples prepared according to the guidelines were submitted to Eurofins Genomics for sequencing with primers shown in Table 1.

### 2.5. Virus Stock Preparation and Titration

Mutant VACV selected after screening was used to infect confluent Vero monolayers in six-well plates. After 1 h of adsorption, the residual inoculum was removed and replaced with DMEM supplemented with 10% FCS and 1% agarose. The plates were incubated for 48 to 72 h at 37 °C in 5% CO_2_. An individual viral plaque was picked and used to infect Vero cells as described. Clonal isolates were obtained after another two rounds of plaque purification from infected cells grown in agarose-containing medium. After the third round of plaque purification, a virus stock was prepared by infecting Vero cells (4 × 10^6^ cells in one 25 cm^2^ culture flask) with the purified mutant VACV. After three freeze–thaw cycles, the lysed cell suspension was centrifuged at 1200× *g* for 10 min. Sanger sequencing was again performed on the whole E9L gene to ensure the presence of the mutations. Virus yield was determined in 96-well plates of Vero cells. Titration was performed using serial fourfold dilutions of the samples, and the viral titer was expressed in TCID_50_/_mL_ [16]. Virus stocks were conserved in 200 µL aliquots at −80 °C.

### 2.6. Plaque Reduction Assay

Confluent Vero cells seeded in six-well plates were inoculated with 0.8 mL virus suspension containing ~100 PFU. After adsorption for 1 h at 37 °C, the medium was removed and the cells overlaid with 0.8 mL DMEM and 2% FCS containing the appropriate concentration of the drug cidofovir (CDV). After 1 h at 37 °C, 1.6% carboxymethyl cellulose (VWR, Rosny-sous-bois, France) diluted in DMEM and 2% FCS was added. The plates were incubated for 3 days at 37 °C in a 5% CO_2_ incubator. Monolayers were fixed and stained in 3.7% formaldehyde, 0.1% crystal violet, and 1.5% methanol. The plaques were counted microscopically.

### 2.7. Growth Kinetics of Mutant VACV In Vitro

Twenty-four-well plates containing 8 × 10^5^ Vero cells were infected with VACV at an MOI of 0.05 in DMEM at 37 °C in a 5% CO_2_ atmosphere. At 1 h post infection (hpi), fresh medium containing 5% *v*/*v* FCS (2.5% final concentration) was added, and the plates were incubated at 37 °C. Cells were harvested at 8, 24, 48, and 72 hpi. Virus yield was determined by titration of the virus, as described above.

### 2.8. Statistical Analysis

Raw data were analyzed using R (4.0.3) and RStudio (1.4.1103) software. The packages used were stats, coin, multcomp, doBy, boot, and ggplot2. Statistical comparisons between groups for a single quantitative variable were performed using resampling tests for intergroup comparisons, paired by experiment. *p*-Values were corrected for the multiplicity of tests using the Benjamini/Hochberg method, and *p* = 0.05 was considered statistically significant; tests were bilateral. Data are expressed as means with bootstrapped 95% confidence intervals.

## 3. Results

### 3.1. Selection of Recombinant VACV Carrying a Silent Mutation in E9L Using CRISPR/Cas9

Two steps are crucial for the generation of recombinant VACV carrying an exogenous gene. First, the desired gene must be integrated into the viral genome by homologous recombination, and second, a specific selection tool must be applied to isolate modified viruses from parent viruses. In this context, the contribution of CRISPR/Cas9 technology to editing the VACV genome is interesting because it allows genetic material to be integrated at a precise location in the genome while applying, at the same time, selective pressure on the VACVs. Indeed, experiments have shown that the Cas9-induced DSBs of the viral genome impair replication [14]. Thus, recombinant viruses lacking the gRNA target sequence are positively selected from the large background of wildtype (WT) viruses, for which the genome remains sensitive to Cas9 cleavage. To produce mutant VACVs carrying point mutations in the E9L gene (encoding the viral DNA polymerase), we used the CRISPR/Cas9 system. However, in this case, the specific selection of mutant VACV could not be achieved, since the gRNA targeted sequence is still present in the recombinant genome (i.e., both the WT and mutant genomes are sensitive to Cas9 cleavage). To overcome this hurdle, we modified the CRISPR/Cas9 system by introducing a critical silent mutation in the E9L gene cloned into the donor vector. Three gRNAs targeting different regions of E9L were designed (Table 2). For each construct, the PAM motif (NGG) downstream of the DNA sequence targeted by the gRNA was mutated at the second nucleotide. As the presence of the PAM sequence is strictly required for gRNA recognition and cleavage [17], it is expected that recombinant VACV genomes bearing this mutation would be insensitive to Cas9 nuclease. Furthermore, the mutation is located in a restriction site, which will result in distinct cleavage patterns of WT and mutant E9L after restriction enzyme digestion, and thus help in the rapid identification of mutant VACVs.

We tested whether recombinant VACV carrying the silent mutation could be easily rescued using our modified CRISPR/Cas9 system, by infecting CV-1 cells with WT VACV and electroporating them with plasmids encoding Cas9, one gRNA, and the corresponding E9L donor vector (Figure 1). At three days post-infection, the viruses were recovered and diluted, such that individual plaques could be easily picked on Vero cells. After viral DNA extraction, a 2415-bp-long fragment of genomic E9L was selectively amplified by PCR (Table 1) and digested with BspEI or MmeI. Distinct cleavage patterns were obtained after endonuclease digestion, indicating that mutant VACVs with the silent mutation in the E9L gene could be readily identified (Figure 2A). We then wanted to determine whether the CRISPR/Cas9 system, as used in our experiments, allows for better selection of mutant VACVs relative to the traditional method based on HR events that occur between the VACV genome and homologous sequences in the shuttle vector. We performed five independent experiments, in which mutant VACV carrying the silent mutations 867, 1533, and 1695 were produced using CRISPR/Cas9 (Figure 2B). As a control, mutant VACV 867 was rescued using the conventional method (i.e., without Cas9 and gRNA). For each mutant, 15 to 20 plaques were isolated, the viral genome extracted, and restriction pattern analysis was performed on the PCR-amplified E9L fragment. When mutant VACV was produced without the use of CRISPR/Cas9 (Ctrl-867), only 14% of the isolated VACV carried the 867 silent mutation. On the contrary, mutant VACV recovery was significantly higher (*p* < 0.05) when using CRISPR/Cas9: 70% for VACV 867, 85% for VACV 1695, and 95% for VACV 1533. These data clearly show that the modified CRISPR/Cas9 system developed in the present study allows for the easy selection and identification of VACV bearing a point mutation.

### 3.2. Rapid Generation of Recombinant VACV with a Resistance Phenotype to Cidofovir

As the viruses produced above were WT viruses, we assessed whether we could add other mutations in E9L to recover VACV with distinct phenotypes. As a proof of concept, we first produced a VACV with a resistance phenotype to cidofovir (CDV). As described in earlier studies, two mutations in E9L (A314T and/or A684V) can confer resistance to CDV, although the degree of resistance is significantly higher when the virus encodes both mutations [18]. We introduced the two mutations, A314T and A684V, in the E9L donor vector bearing the silent mutation 1695. The recombinant VACV was rescued, and restriction pattern analysis was performed on 10 individual plaques. All the isolated viruses carried the silent mutation, indicating, again, the efficiency of the method described (Table 3). We then sequenced the complete E9L gene from each isolated VACV. Interestingly, we obtained a number of viruses with only the A314T mutation (2/10) or A684V mutation (1/10). Two VACVs had neither of the two mutations. However, 5/10 recombinant VACVs encoded the two expected mutations. We further characterized the phenotype of one of these viruses (named VACV CDV^R^) in a plaque reduction assay (Figure 3). We did not observe inhibition of VACV CDV^R^ plaque formation when infected cells were treated with up to 200 µM CDV. By contrast, VACV 1695 was clearly sensitive to CDV at a concentration as low as 50 µM. These results demonstrate the resistance phenotype of VACV CDV^R^ toward CDV and the WT phenotype of VACV 1695. Indeed, earlier studies showed the EC_50_ of CDV on WT VACV to be approximately 50 µM [19,20].

### 3.3. Production of VACV with Mutations in the Domain of E9 Interacting with A20

Earlier structural experiments from our laboratory identified a domain within E9 that serves as the binding site for the processivity factor subunit A20 [21]. A detailed structure of the interface obtained by NMR showed that the main contacts involve conserved residues, such as E9:L578 and E9:I582, from the E9 insert 3 helix [22]. Biochemical data showed that mutation of both residues prevents E9/A20 binding [21]. Thus, we tested whether a version of VACV with alanine substitutions at positions E9:L578 and E9:I582 would be replication competent and rescuable. The two mutations L578A and I582A were introduced into the E9L donor vector bearing the silent mutation 1695. As a control, we also built a donor vector, in which E9:L588 and E9:Q589 were mutated to serine and alanine, respectively. These residues, although in close proximity to E9:L578 and E9:I582, are not in contact with A20 [22], and these mutations do not prevent the E9/A20 interaction [21]. Restriction pattern analysis was performed on 10 isolated plaques for each virus. For VACV^L588S+Q589A^, 10/10 plaques corresponded to recombinant viruses (Table 3). Sequencing of E9 further confirmed that all the rescued viruses carried the expected mutations. For VACV^L578A + I582A^, 7/10 clones carried the silent mutation, but none had the L578A or I582A substitutions. We analyzed another 25 plaques, and none of the 20 recombinant VACVs for which E9L was sequenced had the expected L578A or I582A mutations (Table 3). Thus, our failure in rescuing VACV^L578A + I582A^ highlights the critical features of these residues in the E9/A20 interaction, and the importance of this interaction for viral genome synthesis.

Finally, we tested whether we could rescue a VACV carrying the single E9 L578A mutation. Indeed, the central residue E9:L578 at the N-terminal end of the helix fits into the hydrophobic pocket localized at the C-terminal end of A20 [22]. Thus, we introduced a triple-nucleotide substitution into the E9L donor vector in which residue L578 (TTG) was changed to A578 (GCA). Interestingly, among the 10 plaques analyzed, four recombinant VACVs carried the E9 A578 mutation (Table 3). We further characterized this mutant VACV to determine whether the mutation E9 L578A had an effect on viral replication. We assessed the growth kinetics of VACV^L578A^ in Vero cells, and compared them to those of VACV 1695 and VACV^L588S + Q589A^. There were no significant differences in virus production between the three viruses (Figure 4), indicating that the single L578A mutation did not impair VACV replication in vitro. Furthermore, the four isolated clones of VACV^L578A^ were passaged 10 times, and none displayed reversion at residue 578. Overall, our results indicate that mutation of both E9L578 and I582 is critical for VACV replication. However, the single mutation E9 L578A is tolerated.

## 4. Discussion

The development of CRISPR/Cas9 technology has revolutionized research on genome manipulation of eukaryotic cells. Most recently, this system has been successfully used for the genetic engineering of large DNA viruses, such as herpes simplex virus, adenovirus, and VACV. Thus, in the future, the CRISPR/Cas9 tool will likely be widely used in the oncolytic virus field, and for the production of vaccine candidates against infectious diseases [23].

VACV has long been used for the expression of exogenous genes because it can accommodate large segments of foreign genetic material, and it is easy to genetically modify by homologous recombination. However, this activity is of limited efficiency, and the pool of recombinant viruses must be enriched using a selectable marker incorporated into their genome. The first studies employing the CRISPR/Cas9 system showed a significant increase in the ability to select recombinant viruses relative to the conventional homologous recombination procedure [11,12]. The authors argued that Cas9/gRNA-induced DSBs of the viral genome increased the efficiency of homologous recombination, sometimes by up to 94% [12,23]. However, a more recent study by Gowripalan et al. did not report any improvement in the basal rate of homologous recombination upon VACV genome cleavage. Instead, they observed that the efficient cleavage of the genome by Cas9 (regardless of whether the targeted gene is essential or not) leads to the inhibition of viral DNA synthesis, thus applying positive selection for the recombinant VACV [14].

As described above, the CRISPR/Cas9 tool is well suited for the integration of DNA fragments into the VACV genome, but the selection of recombinant viruses bearing single mutations in a specific gene is still challenging, particularly if the mutant virus displays a weak phenotype. Recently, Laudermilch and Chandran developed a new strategy to generate recombinant VACVs, including mutants with point mutations [24]. Referred to as MAVERICC, this approach relies, first, on the in vitro cleavage of purified VACV DNA by Cas9/gRNA. The cleaved DNA is then co-transfected with the desired amplicon into cells previously infected with the fowlpox virus as a helper virus. Within the cell, trimolecular recombination takes place, and recombinant VACVs are produced. Here, we propose an alternative method, in which the Cas9/gRNA-induced VACV genome cleavage occurs within the cell. Our system is based on the introduction of a mutation in the PAM sequence adjacent to the designed gRNA. This mutation, which results in no amino-acid change, allows for easy selection and screening of mutant VACVs. This method can be used to insert point mutations almost anywhere in the VACV genome. Indeed, nearly 9000 PAM sequences (NGG) have been found throughout the VACV genome by bioinformatics analysis [11], and numerous restriction enzymes that recognize and cleave GG-containing restriction sites are available. The silent substitution of one of the Gs results in the loss of the restriction site. The possibility of integrating point mutations in the VACV genome (and more generally in the poxvirus genome) is of major interest in the determination of the precise structure/function of viral proteins.

Over the last several years, our laboratory has been interested in understanding how essential VACV proteins involved in genome replication are organized at the replication fork. We have focused, in particular, on the DNA polymerase holoenzyme E9/A20/D4, E9 being the catalytic subunit of DNA polymerase, and A20/D4 the heterodimeric processivity factor [25,26]. We recently solved the high-resolution structure of E9 [21], an enzyme that has been well characterized through a number of genetic and biochemical studies [27]. It is a member of the DNA polymerase family B [28], with DNA polymerase and 3′–5′ proofreading exonuclease activities [29]; interestingly, it was also shown to catalyze the annealing of single-stranded DNA [8], an activity not found in other B family DNA polymerases. From a structural point of view, three poxvirus-specific inserts were observed on E9. Although the functions of insert 1 (aa 208–233) and insert 2 (aa 354–434) have not yet been elucidated, insert 3 (aa 567–617), located in the palm domain, has been shown to interact with the A20 processivity factor subunit [21,22]. With this data in hand, the ability to determine the phenotype of mutant VACV with mutations in E9L is of great interest. Thus, we were able to target residues that were previously shown to be critical for the E9/A20 interaction using our modified CRISPR/Cas9 system [21,22]. Our failure to rescue VACV ^L578A + I582A^ confirms the importance of these two residues in maintaining the integrity of the VACV DNA polymerase holoenzyme. The fact that VACV ^L578A^ could be produced also confirms what we observed in our structural studies, that a number of other hydrophobic residues of A20 also establish contact with E9 [22], which, in this case, maintained the E9/A20 interaction. It is also noteworthy that replacing L578 with Ala (another nonpolar residue) may not have a strong mutational effect.

In the future, we plan to investigate the function of VACV-specific inserts 1 and 2 by introducing targeted mutations in these domains. Insert 1 forms an α-helix that is exposed at the surface of E9, and possibly involved in an interaction with an unknown protein. Insert 2, located within the exonuclease domain, is intriguing, and may be involved in the unique role of E9 in recombination [8]. However, poxvirus DNA replication and recombination appear to be tightly linked processes and, to date, no mutants with functionally separate activities have been identified.

## Figures and Tables

**Figure 1 viruses-14-01559-f001:**
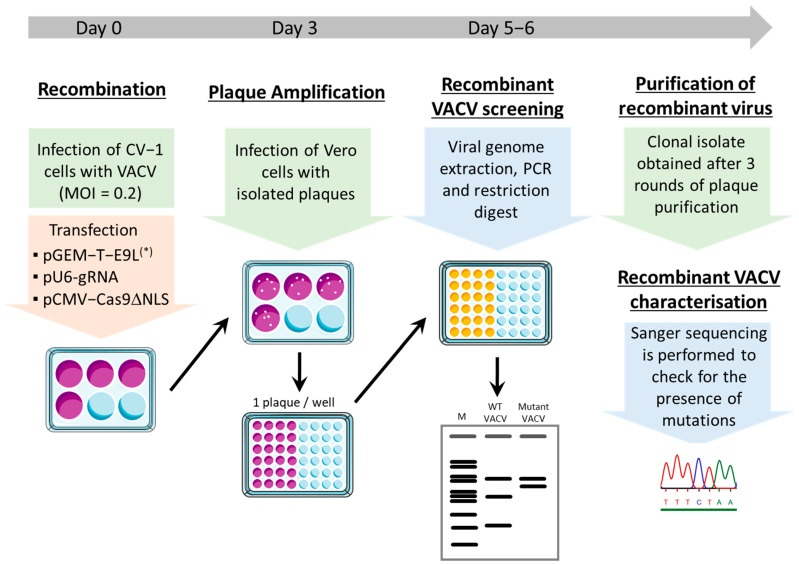
Overview of the rapid generation of point mutations in the VACV genome using Crispr/Cas9, showing the steps leading to mutant VACV selection. Recombination takes place in CV-1 cells, where VACV genome cleavage is induced by Cas9/gRNA. (*) E9L carries a silent mutation in the PAM motif adjacent to the DNA sequence targeted by the gRNA, and eventually other mutations of interest (see text). VACV plaques are amplified, and restriction pattern analysis of E9L is performed. Mutant viruses are then plaque-purified and further characterized by sequencing.

**Figure 2 viruses-14-01559-f002:**
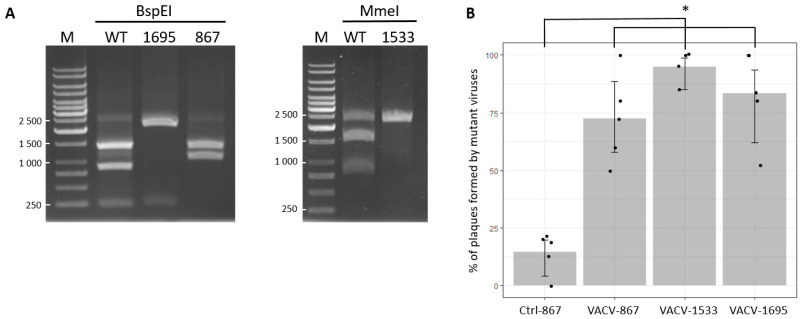
Efficient mutagenesis of the VACV genome using CRISPR/Cas9. (**A**) Agarose gel electrophoretic restriction fragment patterns obtained after PCR amplification of a 2415-bp-long fragment of E9L (Table 1) and digestion with BspEI or MmeI. BspEI digestion of WT E9L produced three fragments (1361 bp, 828 bp, and 226 bp). Two fragments of 2189 bp and 226 bp were obtained for E9L 1695, as well as for E9L 867 (1361 bp and 1054 bp). The 2415-nucleotide-long WT E9L PCR was only partially digested by MmeI, and produced three fragments of 1548 bp, 777 bp, and 90 bp (not shown). Two fragments of 2325 bp and 90 bp (not shown) were obtained for E9L 1533. Lane M: DNA size standard (GeneRuler 1 kb, Thermo Fisher Scientific). The sizes of the relevant bands are indicated in bp. (**B**) Percentage of plaques formed by mutant viruses. Mutant viruses were identified by restriction pattern analysis, as shown in panel (**A**). Results are presented from five independent experiments (except for mutant 1533, four experiments) for which 15/20 plaques were isolated and analyzed as described above (data are shown as the mean ± standard deviation). * Indicates *p* < 0.05 compared to Ctrl-867 group (using resampling tests for intergroup comparisons).

**Figure 3 viruses-14-01559-f003:**
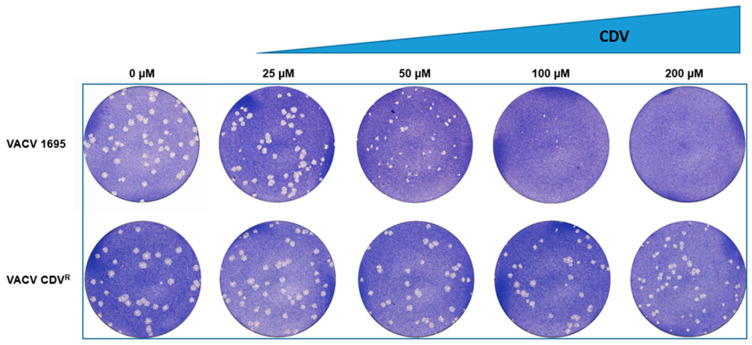
Generation of cidofovir-resistant VACV. VACV 1695 and VACV CDV^R^ (carrying both A314T and A684V mutations) were tested for plaque formation on Vero cell monolayers overlaid with medium containing increasing concentrations of CDV. Three days post-infection, cells were fixed and stained with 0.1% crystal violet. Photographs of the crystal-violet-stained cell monolayers are shown.

**Figure 4 viruses-14-01559-f004:**
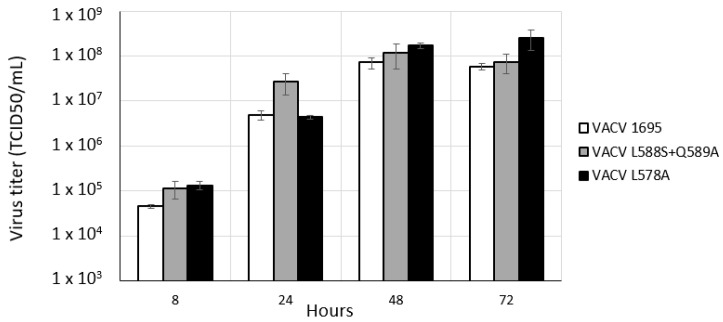
Growth properties of E9L mutant viruses. Vero cells were infected with the indicated viruses at an MOI of 0.05 and cultured at 37 °C. Virus yield was determined at each of the indicated time points, and is expressed in TCID_50_/_mL_. Results are expressed as the mean ± standard deviation for three experiments.

**Table 1 viruses-14-01559-t001:** Oligonucleotide primers used in this study.

Primer Use and Name	Oligonucleotide Sequence (5′-3′) *
**E9L cloning in pGEM-T**	
E9L-F	CTATGGTACCATGGATGTTCGGTGCATTAATTGGTT
E9L-R	CTATAAGCTTTTATGCTTCGTAAAATGTAGGTTTTGAAC
**pGEM-T-E9L mutagenesis**	
E9L-1695-R	GAAACATTAGTCGGTGTCG
E9L-1695-F	**G**GGAGATAGATTTCCAAAG
E9L-1533-R	GAAACTAAAACTATCTTAGTTAGATC
E9L-1533-F	**T**AACAATAGCTTTAACAGTGG
E9L-867-R	GACAAAAAGGAAGCTGTAC
E9L-867-F	**G**GGAGATCTAAAGATAATCTTC
E9-L578A-I582A-R	CTTCC**GC**TCTATTGGTACTAACAAC
E9-L578A-I582A-F	AAGAA**GC**AAATAATCAGCTATTGCTTC
E9-A314T-R	TAGGCGGCATG**A**CCAATACTAC
E9-A314T-F	CTCCCTTATGACTAGACTGATTTCTC
E9-A684V-R	**A**CCGAAGCGTATGAGTATAGAGC
E9-A684V-F	TAAGAGTTGCACATCCATAGGACG
E9-L588S-Q589A-R	A**GA**CAATAGCTGATTATTTATTTCTTCTTC
E9-L588S-Q589A-F	**GC**GAAATATCCACCTCCTAGATATATTAC
E9-L578A-R	**C**TCTATTGGTACTAACAACG
E9-L578A-F	**CA**GAAGAAGAAATAAATAATCAGC
**pCMV-Cas9 mutagenesis**	
Cas9ΔNLS-F	TGATGACTCGAGTCTAGAGGGCCCG
Cas9ΔNLS-R	GTCGCCTCCCAGCTGAGACAGGTCG
**PCR screening for rVACV ^†^**	
E9-screen-F	CTAACAAAGAGCGACGTACAAC
E9-screen-R	GAAGCCGTCGATAGAGGATG
**E9L sequencing**	
E9-seq-F	CTAACAAAGAGCGACGTACAAC
E9-seq-R	CTTCCCCAATGTTTGGGATTC
E9-790-F	CGTTAAAGGTAACGACGTAG
E9-643-R	GAAGCCGTCGATAGAGGATG
E9-2262-F	GGAGTCGGTATCTCCATACAC
E9-2100-R	GGTACTAAATGGAGCAGAG

* Bases in bold capital letters are different from the wildtype sequence. ^†^ rVACV: recombinant vaccinia virus.

**Table 2 viruses-14-01559-t002:** gRNA targets on E9L.

**gRNA867**	**3’-CTCTTCTAATAGAAATCTAGAGGCCTGTT-5’** **5’-GAGAAGATTATCTTTAGATCTCCGGACAA-3’** ** | ** ** C(867)**
**gRNA1533**	**3’-** **CCAGGTGACAATTTCGATAA** CAACCT **TTG** **-5’** **5’-GGTCCACTGTTAAAGCTATTGTTGGAAAC-3’** ** | ** ** A(1533)**
**gRNA1695**	**3’-****ACATAGAAACCTTTAGATAG**AGGCCTTTG**-5’** **5’-**TGTATCTTTGGAAATCTATCTCCGGAAAC**-3’** ** | ** ** C(1695)**

The E9L sequences targeted by the gRNAs are shown. The gRNA and PAM sequences are shown in black and red capital letters, respectively. The BspEI (5′TCCGGA3′) and MmeI (5′TCCRAC(N)203′) restriction sites are underlined. For each construct, the silent mutation is indicated together with its position within the E9L gene (3018 nucleotides).

**Table 3 viruses-14-01559-t003:** Analysis of mutant VACV.

Virus Name	Nb of Positive Clones *	Nb of Mutant VACV with Expected Mutations ^†^
VACV CDV^R^	10/10	2/10: no mutation 2/10: A314T only 1/10: A684V only 5/10: A314T + A684V
VACV ^L578A + I582A^	7/10 20/25	0/7 0/20
VACV ^L588S + Q589A^	10/10	10/10
VACV ^L578A^	7/10	4/7

* After cleavage pattern analysis. ^†^ After E9L sequencing.

## Data Availability

Not applicable.

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
