# Peer review of "Efficient Method for Generating Point Mutations in the Vaccinia Virus Genome Using CRISPR/Cas9"

_viruses, 2022, doi:10.3390/v14071559_

Round 1

Reviewer 1 Report

In this study, the authors developed a modified version of CRISPR/Cas9 system to facilitate rapid selection and identification of recombinant vaccinia (VV) bearing point mutations from the parental viruses. Additionally, the authors also produced some recombinant VVs with lesions in the E9L gene. As the conventional recombination required selectable markers inserted to the gene of interest, which is unacceptable in therapeutics and vaccines, this results contribute to the generation of marker-free viruses and facilitate the development of VV-based vaccines. 

Nevertheless, there are several points that should be discussed and revised in this study:

1. The results in this study should be divided in different parts with different titles and clear purposes. Do the authors think the experiments at the end (from line 257) supports the overall objective of this study?

2. The methods used in this study should be presented in details:

- How to pick plaques and extract viral genome from plaques? Condition of experiments should be included.

- How to prepare samples for Sanger sequencing and how to analyze?

3. The authors should make schematics for each section/experiments so that the readers will find it easy to follow. For examples, how to create/select/identify recombinant VVs.

4. The figure legends should be presented in details with experiment conditions. There is no information of statistics in the figure legends.

5. The figures 1A and 1B should be carefully prepared (exclude red lines under letters). The representative pictures of plaque assay should be included in figure 1B to show the difference in the number of plaques (along with the graph). In figure 1B, there should be CON group for each mutant VV (not only VV-867).

Despite the fact that the authors established a modifided version of CRISPR/Cas9 for easy selection and identification of VVs, I think the data and presentation of this study is not good enough for publication.  

Author Response

Reviewer 1

  1. The results in this study should be divided in different parts with different titles and clear purposes. Do the authors think the experiments at the end (from line 257) supports the overall objective of this study?

Headings were added in the results section. The overall objective of the study is to develop a technology allowing for rapid selection of recombinant VACV carrying single mutations. Once the protocol was established, the system was applied to confirm the importance of some residues on E9 that were shown (by us) to be involved in the interaction with A20.  We do believe that these “proof of concept” experiments are important for the study.

  1. The methods used in this study should be presented in details:

- How to pick plaques and extract viral genome from plaques? Condition of experiments should be included.

Picking plaques is a common technique which consists to isolate a viral clone through the overlay agarose using a tip. The agarose plug is transferred onto a new cell culture by pipetting up and down several times. As described in the Materials and Methods section (Screening of recombinant viruses), plaques were amplified on Vero cells seeded in 96 well plates for three days. The viral genome was extracted from the infected cells using the QIAamp DNA mini kit (Qiagen) following the supplier instructions.

- How to prepare samples for Sanger sequencing and how to analyze?

A new section was added in the Materials and methods describing how samples for Sanger sequencing were prepared.

  1. The authors should make schematics for each section/experiments so that the readers will find it easy to follow. For examples, how to create/select/identify recombinant VVs.

We agree with the reviewer. Instead of schematics for each section, we decided to draw a schematic presentation of the whole technique. It is now included in the text as figure 1.

  1. The figure legends should be presented in details with experiment conditions. There is no information of statistics in the figure legends.

We think that in general our legends described the experimental conditions. However, an experimental detail was added in legend of Figure 1. In the same legend information on statistics was also added.

  1. The figures 1A and 1B should be carefully prepared (exclude red lines under letters).

Done

 The representative pictures of plaque assay should be included in figure 1B to show the difference in the number of plaques (along with the graph).

Figure 1B do not represent a difference in the number of plaques but a percentage of mutant viruses, which was determined by restriction pattern analysis (as shown in Figure 1A).  

In figure 1B, there should be CON group for each mutant VV (not only VV-867).

In our experiment, the control is to perform E9L recombination in absence of Cas9/gRNA (i.e. the conventional method). We showed that the efficiently is low as previously reported in the literature. Thus, we felt that it was not necessary to perform this control with the other pasmids encoding E9.

Reviewer 2 Report

The Ms titled ‘Efficient Method for Generating Point Mutations in the Vaccinia Virus Genome Using CRISPR/Cas9’ showed a new method to rapidly generate point mutation in Vaccinia Virus using CRISPR-Cas9 system. It is really interesting idea and potentially can apply to other DNA viruses as well. This methodology will be a very useful tool for Vaccinia gene functional study.

A few points to be addressed to make it easier to understand for the readers:

1. it will be great if a schematic presentation included to describe the method

2. it will be great to have a heading for each result section.

Author Response

Reviewer 2

  1. it will be great if a schematic presentation included to describe the method

We agree with reviewer 2. A schematic presentation of the technique was made and included as figure 1 in the text.

  1. it will be great to have a heading for each result section.

Headings were added in the results section.

Reviewer 3 Report

Recommended for publication

Author Response

Reviewer 3

This reviewer recommended publication without modification.

Round 2

Reviewer 1 Report

In figure legend 1, the authors added “When Cas9/gRNA is used, the results are significantly higher than without Cas9/gRNA (p<0.05)”. This can be excluded from the figure legend as it is a conclusion (should be in the result section). There should be a sign on the graph to show the significant difference between different groups. For examples, “*” and indicate p*<0.05 compared to Ctrl-867 group. How did the authors determine p values? It seems like p-values were determined using the Benjamini/Hochberg method (in statistical analysis section). Even so, I think it should be included in figure legend as well.

It is clear that mutant viruses were determined by restriction pattern analysis, but how did the authors determined the percentage of plaques formed by mutant viruses? I guess it depends on the number of mutant viruses detected by restriction pattern analysis and the isolated plaques (15-20 plaques) at the beginning, is it right?

And since the authors mention “the percentage of mutant virus”, I was wondering if there is a correlation between the “percentage of mutant viruses” and “percentage of plaques formed by mutant viruses”. I am a bit confused of two terms that the authors mentioned.

Author Response

In figure legend 1, the authors added “When Cas9/gRNA is used, the results are significantly higher than without Cas9/gRNA (p<0.05)”. This can be excluded from the figure legend as it is a conclusion (should be in the result section).

The sentence has been removed from figure legend 1 and is present in the result section.

There should be a sign on the graph to show the significant difference between different groups. For examples, “*” and indicate p*<0.05 compared to Ctrl-867 group.

Figure 1 has been modified has suggested by reviewer 1.

How did the authors determine p values? It seems like p-values were determined using the Benjamini/Hochberg method (in statistical analysis section). Even so, I think it should be included in figure legend as well.

As mentioned in the Material and and methods section, “Statistical comparisons between groups on a single quantitative variable were performed using resampling tests for group versus group comparisons, pairing by experiment. P-values were corrected for the multiplicity of tests using the Benjamini/Hochberg method.”

A sentence was also added in the legend of figure 1.

It is clear that mutant viruses were determined by restriction pattern analysis, but how did the authors determined the percentage of plaques formed by mutant viruses? I guess it depends on the number of mutant viruses detected by restriction pattern analysis and the isolated plaques (15-20 plaques) at the beginning, is it right?

Yes it is right.  For each transfection/infection experiments, 15 to 20 plaques were picked and tested by restriction pattern analysis. The percentage of plaques formed by mutant viruses was calculated from these isolated 15 to 20 plaques. For each construct the whole experiment was repeated 4 or 5 times.

And since the authors mention “the percentage of mutant virus”, I was wondering if there is a correlation between the “percentage of mutant viruses” and “percentage of plaques formed by mutant viruses”. I am a bit confused of two terms that the authors mentioned.

Yes, we believe that there is a correlation between the percentage of plaques formed by mutant viruses and the global percentage of mutant viruses since one infectious plaque is formed by one virus.

Reviewer 2 Report

The Ms looks fine to me

Author Response

Reviewer 2 said : The Ms looks fine to me